# Tuning Q-Factor and Perfect Absorption Using Coupled Tamm States on Polarization-Preserving Metasurface

Natalya V. Rudakova [1,2], Rashid G. Bikbaev [1,2], Larisa E. Tyryshkina [1,2], Stepan Ya. Vetrov [1,2] and Ivan V. Timofeev [1,2,*]

1   Kirensky Institute of Physics, Federal Research Center KSC SB RAS, Krasnoyarsk 660036, Russia; natalya-v-rudakova@iph.krasn.ru (N.V.R.); bikbaev@iph.krasn.ru (R.G.B.); letyryshkina@iph.krasn.ru (L.E.T.); stepan-ya-vetrov@iph.krasn.ru (S.Y.V.)
2   Siberian Federal University, Krasnoyarsk 660041, Russia
*   Correspondence: tiv@iph.krasn.ru

**Abstract:** The circular polarization of light flips its handedness after a conventional metallic mirror reflection. Therefore, a polarization-preserving metasurface is a crucially important element in a series of chiral photonic structures. They include tunable cholesteric LCs and anisotropic photonic crystals. Chiral structures are rich in interfacial localized modes including Tamm states. In this report, coupled modes formed as a result of the interaction between two chiral optical Tamm states or a chiral optical Tamm state and a chiral Tamm plasmon polariton are analytically and numerically investigated. It is shown that the effective control of coupled modes can be carried out by changing the pitch of the cholesteric and the angle between the optical axis of the cholesteric and the polarization-preserving anisotropic mirror. The influence of the metasurface period on the spectral characteristics of coupled modes is investigated. The possibility of realizing a bound state in the continuum of the Friedrich–Wintgen type, resulting from the destructive interference of coupled modes, which leads to the collapse of the resonance line corresponding to the chiral optical Tamm state, has been demonstrated.

**Keywords:** bound state in the continuum; optical Tamm state; coupled modes; cholesteric liquid crystal; polarization-preserving metasurface





## 1. Introduction

For the development of modern photonics technologies, it is necessary to design and create new functional elements of optical devices that allow for obtaining the required and controlled spatial spectral and polarization characteristics of radiation in the specified ranges. In 1932, Tamm proposed a theory about the possibility of the existence of special surface localized states of electrons at the surface of a crystal, which were later called "Tamm" states. By analogy with the electronic Tamm state from solid-state physics, in optics, a special type of localized electromagnetic mode excited by the normal incidence of light on a crystal is called the optical Tamm state (OTS). It occurs between two mirrors, during multiple re-reflections, as a result of which the light is localized at the interface of the two media [1,2]. Mirror properties can be exhibited by photonic crystal (PC) structures, metallic and chiral materials, and various metasurfaces. The spatial distribution of the field localized at the OTS frequency exponentially decreases with increasing distance from the interface in both (the transmitting and reflecting) directions. If a thin layer of metal conjugated with a Bragg dielectric reflector is used as one mirror, then a Tamm plasmon polariton (TPP) with a zero wavenumber along the surface can be obtained at its boundary using direct optical excitation [3–7]. The Tamm plasmon polariton, in contrast to the conventional surface plasmon polariton (SPP), if the frequencies are above the light cone, can have both TM and TE polarization [8]. The advantage of the TPP is the high absorption quality factor due to low material losses and the large resonant volume of the Bragg reflector. Further investigation of the properties of optical localized states stems from

their potential for use in various optical devices, including absorbers [9,10], emitters [11,12], sensors [13,14], photodetectors [15–18], organic solar cells [19–22], beam steering [23], and lasers [24–28], especially vertical-cavity lasers [29,30] and filters [31].

A wide range of applications of localized optical states in photonics devices makes the task of controlling their characteristics relevant today. In addition to obtaining the required predicted optimal spectral properties of the structure as a result of the technological process, it is necessary to control the modes, which can be carried out after the fabrication of the structure. It is promising to use chiral media—anisotropic media with a violation of the mirror symmetry of optical properties—having a continuous helical symmetry of the dielectric constant tensor to obtain and control spectral and polarization properties [32–34]. Such materials, for example, a cholesteric liquid crystal (CLC, cholesteric), have an optical axis that rotates in space in the form of a helical spiral, so the band gap appears only for one circular polarization that coincides with the direction of rotation; thus, the CLC is a one-dimensional chiral photonic crystal (PC) with a photonic band gap in the spectrum. Light with a direction opposite to that of the twist circular polarization does not diffract, but passes through the structure unhindered, thus exhibiting selective diffraction reflection with respect to polarization. In addition, CLCs are highly sensitive to changes in electric and magnetic fields and ambient temperature, as well as to doping of the crystal with various micro- and nanoparticles or dye molecules, which leads to the possibility of qualitative restructuring of the spectral characteristics of radiation of a cholesteric liquid crystal. The synergy of active cholesteric and reflective PC yields CLC-PC lasers [35]. The localized surface state that is excited at the interface between a CLC and a chiral plasmonic mirror exponentially decreases with increasing distance from the interface. It is called a chiral Tamm plasmon polariton (CTPP) [2]. Recently, it was experimentally investigated and proposed as a temperature sensor device [36]. The Q-factor can be improved by replacing the plasmonic metasurfaces with a dielectric polarization-preserving anisotropic mirror (PPAM) [37]. As a result, a high-quality chiral optical Tamm state (COTS) [38] is excited at the CLC-PPAM interface. The spectral manifestation of the COTS can be observed both upon reflection and transmission. The high reflectance coefficient from the PPAM is provided by a large number of layers, which complicates the technology of manufacture of these mirrors. Reducing the PPAM layers, while maintaining the Q-factor of the localized state, can be achieved due to its conjugation with the metasurface, as it was demonstrated in the work of [39].

The localized optical state arising at the interface of the media can hybridize with other types of modes existing at the same time in the photonic crystal structure, and this phenomenon is analogous to coupled oscillators. The interaction of states is expressed by the appearance in the transmission and reflection spectra of the quasi-crossing of modes when the position of one of the localized states is rearranged. The presence of hybridization allows for obtaining additional methods for controlling the spectral characteristics and resonant amplitude of modes. It is possible to demonstrate hybrid modes formed by two identical coupled Tamm plasmon polaritons localized at the edges of a photonic crystal adjacent to two layers of a nanocomposite [40] or upon incorporating a thin metal layer into a planar organic microcavity [41]. An example of the interaction of different types of modes is the hybrid state of the TPP and exciton polaritons [42–44], the TPP and surface plasmon polariton [45–47], the TPP and microcavity modes [48], the TPP and chiral optical Tamm state [49], the TPP and magnetic plasmons [50], and the TPP and guided-mode resonance [51].

In this work, the mechanism of occurrence of various types of mode hybridizations as a result of the interaction of two chiral optical Tamm states or a chiral optical Tamm state and a chiral Tamm plasmon polariton is analytically and numerically considered. The methods of effective control of such coupled modes by changing the pitch of the cholesteric, the angle between the optical axis of the cholesteric and the polarization-preserving anisotropic mirror, and the period of the polarization-preserving metasurface are proposed. The possibility of realizing a bound state in the continuum of the Friedrich–Wintgen type,

leading to the collapse of the resonance line corresponding to the chiral optical Tamm state in the case of using the polarization-preserving metasurface in the structure, is shown.

## 2. Description of the Model

The sketch view of the investigated CLC-PPAM-CLC structure is shown in Figure 1. The multilayer PPAM mirror consists of alternating uniaxial dielectric layers with different refractive indices $n_e^p = \sqrt{\varepsilon_e^p}$ and $n_o^p = \sqrt{\varepsilon_o^p}$. Let us describe these layers using different dielectric tensors, vertical $\hat{\varepsilon}_V$ and horisontal $\hat{\varepsilon}_H$, which can be written as

$$\hat{\varepsilon}_V = \begin{pmatrix} 1.7 & 0 & 0 \\ 0 & 1.5 & 0 \\ 0 & 0 & 1.7 \end{pmatrix}, \quad \hat{\varepsilon}_H = \begin{pmatrix} 1.5 & 0 & 0 \\ 0 & 1.7 & 0 \\ 0 & 0 & 1.7 \end{pmatrix}. \tag{1}$$

The optical axes of the PPAM layers are directed along the $x$ (yellow layer) and $y$ (grey layer) axis, respectively (see Figure 1). The number of unit cells (the number of V-H pairs) in the structure is $N$ and the period of the structure is $\Lambda = d_V + d_H$, where $d_V = d_H = 187.5$ nm are the thicknesses of the unit cell layers. The CLC is an optical chiral medium with continuous helical symmetry of the dielectric permittivity tensor. This medium is characterized by helix pitch $p = 750$ nm, cholesteric layer thickness $L = 3$ µm, and ordinary and extraordinary refractive indices $n_e = \sqrt{\varepsilon_e}$ and $n_o = \sqrt{\varepsilon_o}$, respectively. The CLC permittivity and permeability tensors, respectively, are

$$\hat{\varepsilon}(z) = \varepsilon_m \begin{pmatrix} 1 + \delta\cos(qz) & \pm\delta\sin(2qz) & 0 \\ \pm\delta\sin(qz) & 1 - \delta\cos(qz) & 0 \\ 0 & 0 & 1 - \delta \end{pmatrix}, \quad \hat{\mu}(z) = \hat{I}, \tag{2}$$

where $q = 4\pi/p$, $\varepsilon_m = (\varepsilon_e + \varepsilon_o)/2$, and $\delta = (\varepsilon_e - \varepsilon_o)/(\varepsilon_e + \varepsilon_o)$. CLC1 and CLC2 are denoted as cholesterics with a different helix pitch. The investigated structure is placed in a medium with refractive index $n_{av} = (n_o + n_e)/2$.

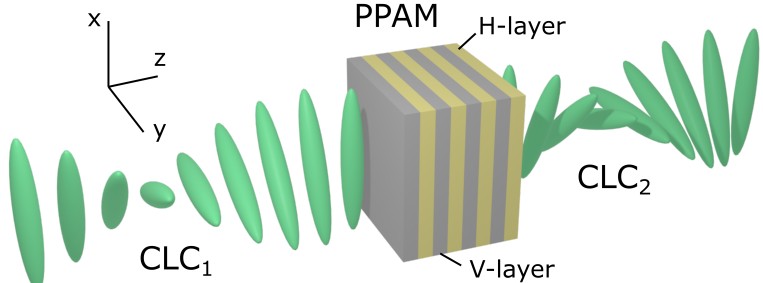

**Figure 1.** Sketch view of the polarization-preserving anisotropic mirror bounded by two cholesteric liquid crystals. CLC$_1$ and CLC$_2$ denote cholesterics with a different helix pitch. H- and V-layers are denoted as layers with optical axes along x and y directions, respectively.

The spectral properties of the CLC-PPAM-CLC structure and the field distribution in it is performed using the Berreman 4 × 4 transfer matrix method [52]. Matrix methods of writing Maxwell's equations are most often used to describe the propagation of light through various optical media with plane-parallel boundaries. In his method, Berreman considers an optical medium in which the parameters are continuously changing. This condition allows you to write Maxwell's equations in the form of a 4 × 4 differential matrix. Increasing the order of the matrix leads to an increase in the accuracy of calculations. The Berreman matrix determines the linear alignment of the tangential components of the electric and magnetic fields at the input of the optical system and at its output. This makes it possible to calculate both the transmission coefficients of a light wave incident on a flat optical system at an arbitrary angle and its reflection coefficients. The Berreman method takes into account multiple interference reflections in the layers of the structure.

From the point of view of optics, liquid cholesteric crystals are an anisotropic birefringent medium with continuous torsion, that is, the direction of the long axes of the molecules in each subsequent layer, consisting of molecules oriented in parallel and freely moving in two directions, forms a certain angle with the direction of the axes of the molecules of the previous layer. In this case, a spiral is formed, the step of which depends on the nature of the molecules and external influences on the cholesteric. When calculating the spectral properties, an inhomogeneous layer of a cholesteric liquid crystal is divided into a sufficiently large number of plane-parallel thin homogeneous sublayers, each of which is approximately homogeneous, and the orientation direction of CLC molecules is strictly set in it. The distribution matrix of the cholesteric in accordance with the Berreman method can be found by sequentially multiplying the distribution matrices of each subsequent separation layer by each other. Then, the equation describing the propagation of light at frequency $\omega$ along the $z$ axis normal to the structural layers has the following form:

$$\frac{d\psi}{dz} = \frac{i\omega}{c}\Delta(z)\psi(z),\tag{3}$$

where $\psi(z) = (E_x, H_y, E_y, -H_x)^T$ and $\Delta(z)$ is the Berreman matrix, which depends on the dielectric function and the incident wave vector.

The optical properties of the CLC-PPAM-metasurface structures were calculated using the Finite-Difference Time-Domain (FDTD) method. The investigated structure was illuminated by the circularly polarized wave with normal incidence along the $z$ axis. Reflectance $R$ was calculated on the right of the simulation box. Periodic boundary conditions were applied at the lateral boundaries of the simulation box, while perfectly matched layer (PML) boundary conditions were used on the remaining right and left sides. An adaptive mesh was used to accurately reproduce the CLC shape. Extensive convergence tests for each set of parameters were performed to avoid undesired reflections on the PMLs.

## 3. Results and Discussion

### 3.1. Coupled COTS-COTS Modes

Reflectance spectra of the structure at different values $N$ are shown in Figure 2.

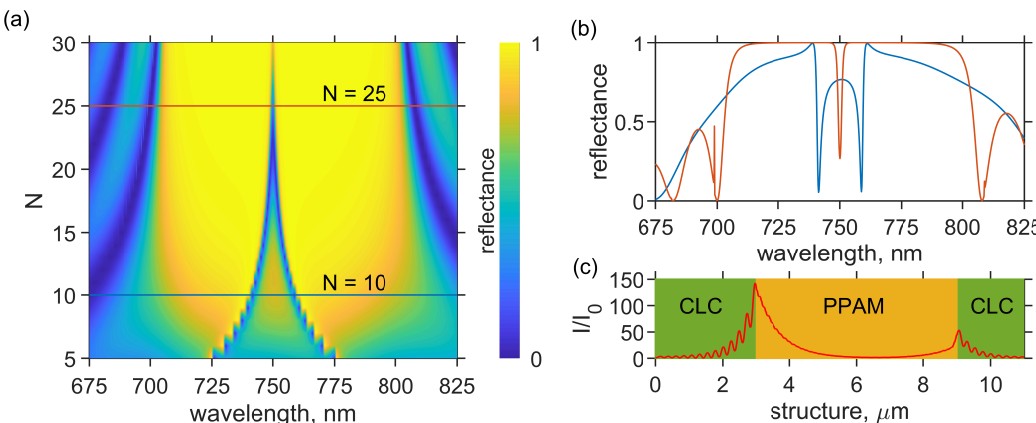

**Figure 2.** (**a**) Reflectance spectra of the CLC-PPAM-CLC structure for different numbers of the PPAM periods $N$. (**b**) Reflectance spectra of the structure for $N = 10$ (blue line) and $N = 25$ (red line). (**c**) Field distribution at the COTS wavelength $\lambda = 750$ nm.

It can be seen that at $N > 20$, only one resonant line is observed in the reflectance spectra. This line corresponds to the chiral optical Tamm state localized at the interface between the cholesteric and multilayer polarization-preserving anisotropic mirror (see Figure 2c). A decrease in the PPAM period number leads to a decrease in reflection within its band gap and, as a consequence, the incident radiation tunnels through it to the second interface with the CLC. As a result, a second COTS is excited at the PPAM-CLC boundary.

The states localized at the PPAM boundaries begin to overlap with each other, which leads to the removal of degeneration and splitting of the resonant line. As a result, two reflection minima in the reflection spectrum of the structure corresponding to the coupled COTS are observed (please see blue line in Figure 2b). Let us denote the localized state at the CLC-PPAM interface as $COTS_1$ and the localized state at the PPAM-CLC interface as $COTS_2$.

As was mentioned above, an important advantage of the CLCs over other types of photonic crystals is their high sensitivity to external fields. A strong dependence of the helix pitch, for example, on temperature or applied voltage can be used to effectively control the COTS splitting value. For this reason, reflectance spectra of the structure for different values of the second CLC pitch were calculated. The results are shown in Figure 3.

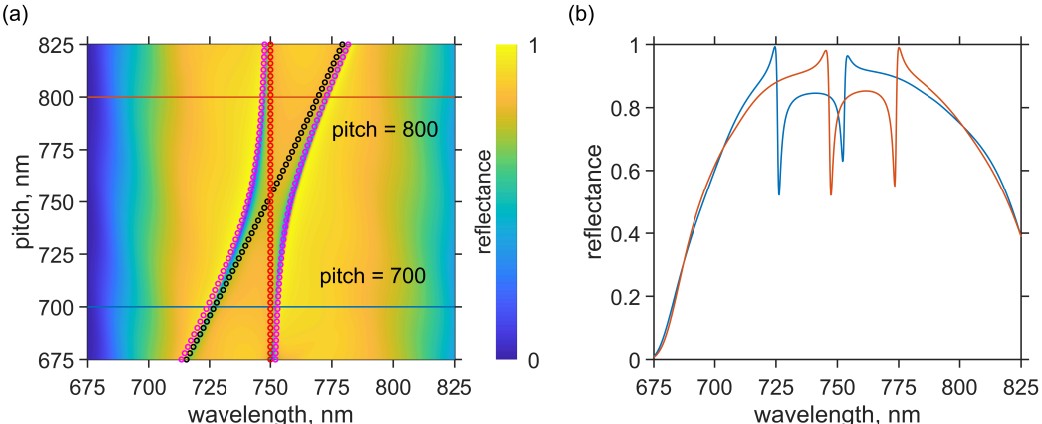

**Figure 3.** (**a**) Reflectance spectra of the CLC-PPAM-CLC structure for different values of the CLC pitch. The eigenfrequency $\omega_{COTS_1}$ is presented as red circles. The analytical solutions of Equation (8) for the eigenfrequency $\omega_{COTS_2}$ are presented as black circles. The analytical solutions for coupled COTS modes obtained by the coupled oscillator model according to Equation (4) are presented as magenta circles. (**b**) Reflectance spectra of the structure for pitch = 700 nm (blue line) and pitch = 800 nm (red line).

It can be seen from the figure that an increase in the CLC pitch from 675 nm to 825 nm leads to a shift in the $COTS_2$ wavelength from 715 nm to 781 nm, while the wavelength of $COTS_1$ does not change. As a result, the avoidance of crossing of resonant lines is observed. The minimum splitting value is achieved at a helix pitch of 750 nm, while a decrease or increase in this value leads to an increase in the splitting value. For helix pitch values equal to 700 and 800 nm, the splitting value is 26 nm (please see Figure 3b).

A similar effect of the resonant mode splitting can be achieved by changing the angle between the optical axis of the PPAM and the second CLC. In this case, the wavelength of the second COTS undergoes a blue shift, while the $COTS_1$ wavelength does not change. Reflectance spectra of the structure for different $\phi$ are shown in Figure 4b.

Such behavior of resonant modes for different values of the CLC pitch and angle $\varphi$ can be explained using the coupled oscillator model, and accordingly, the frequencies of coupled modes can be found by equating the determinant of the matrix to zero:

$$\begin{vmatrix} \omega - \omega_{COTS_1} & \Omega_{COTS_{12}} \\ \Omega_{COTS_{12}} & \omega - \omega_{COTS_2} \end{vmatrix} = 0, \tag{4}$$

where $\omega_{COTS_1}$ and $\omega_{COTS_2}$ are eigenfrequencies of the $COTS_1$ and $COTS_2$, respectively. In our case, $\omega_{COTS_1}$ is constant and equal to 1.33 $\mu m^{-1}$. $\Omega_{COTS_{12}}$ is a coupling coefficient between the modes, which can be estimated from simulation results as the splitting value between the coupled modes.

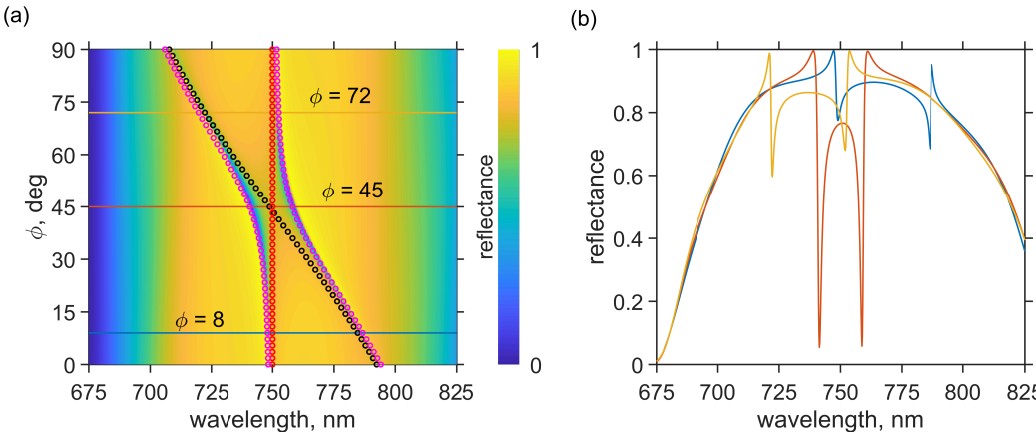

**Figure 4.** (**a**) Reflectance spectra of the CLC-PPAM-CLC structure for different value angles $\varphi$ at the PPAM-CLC interface. The eigenfrequency $\omega_{COTS_1}$ is presented as red circles. The analytical solutions of Equation (8) for the eigenfrequency $\omega_{COTS_2}$ are presented as black circles. The analytical solutions for coupled COTS modes obtained by the coupled oscillator model according to Equation (4) are presented as magenta circles. (**b**) Reflectance spectra of the structure for $\varphi = 8$ deg (blue line), $\varphi = 45$ deg (red line), and $\varphi = 72$ deg (orange line).

As a result, the resonant frequencies of the modes can be expressed as [53]

$$\omega_{\pm} = \frac{\omega_{COTS_1} + \omega_{COTS_2}}{2} \pm \frac{\sqrt{(\omega_{COTS_1} - \omega_{COTS_2})^2 + \Omega_{COTS_{12}}^2}}{2}. \tag{5}$$

The eigenfrequency $\omega_{COTS_2}$ can be determined by the phase matching condition [3]. For conventional Tamm plasmon polaritons, this condition has the following form:

$$|r_m| e^{i\varphi_m} |r_{PhC}| e^{i\varphi_{PhC}} = 1, \tag{6}$$

where $\varphi_m$, $r_m$ and $\varphi_{PhC}$, $r_{PhC}$ are the phases and amplitudes of waves reflected from the metallic layer and photonic crystal, respectively. This is equivalent to the conditions for amplitudes:

$$|r_{PhC}| |r_{PhC}| = 1$$

and phases:

$$\varphi_m + \varphi_{PhC} = \pi n, \tag{7}$$

where $n = 0, 1, 2 \ldots$

For the proposed structure, an additional term $\phi$ will appear in the phase matching condition. During a cycle of two re-reflections, there appears the geometric component of the phase equal to $2\phi$ along with the dynamic phase. The dynamic phase grows with optical distance. Therefore, the phase variation by the $\phi$ angle is not a dynamic phase, as it does not change the optical distance. Such a phase shift is called the geometric phase [54,55]. As a result, the phase matching condition for COTS can be written as

$$\varphi^p + \varphi^c + \phi = \pi n, \tag{8}$$

where $\varphi^p$ and $\varphi^c$ are the reflection phases from the PPAM and CLC, respectively.

The PPAM is a periodically layered medium; thus, it can be described analytically [56]. The expression for the amplitude reflection coefficient is

$$r_N^p = \frac{CU_{N-1}}{AU_{N-1} - U_{N-2}} = \frac{C}{A - \sin(N-1)K\Lambda^p / \sin NK\Lambda^p}, \tag{9}$$

$$\arg r_N^p = \varphi^p, \tag{10}$$

where $U_N = \sin(N+1)K\Lambda^p / \sin K\Lambda^p$ and $K = [1/\Lambda^p]\arccos[(A+D)/2]$ is the Bloch wavenumber.

The elements of the transformation matrix for a cell showing the relation between the amplitudes of the plane waves in the first layer and the respective amplitudes in the neighboring unit cell are as follows:

$$A = e^{ik_{1z}a}\left[\cos k_{2z}b + \frac{1}{2}i\left(q+\frac{1}{q}\right)\sin k_{2z}b\right];$$

$$C = e^{ik_{1z}a}\left[-\frac{1}{2}i\left(q-\frac{1}{q}\right)\sin k_{2z}b\right]; \tag{11}$$

$$D = e^{-ik_{1z}a}\left[\cos k_{2z}b - \frac{1}{2}i\left(q+\frac{1}{q}\right)\sin k_{2z}b\right],$$

where $k_{1z} = (\omega/c)n_e^p$ and $k_{2z} = (\omega/c)n_o^p$ are the wave vectors for the first and second media, respectively; $q = k_{2z}/k_{1z} = \sqrt{\varepsilon_o^p/\varepsilon_e^p}$ is the geometric progression denominator; $\bar{\varepsilon}^p = \left(\varepsilon_o^p + \varepsilon_e^p\right)/2$ is the arithmetic mean over ordinary and extraordinary permittivities for the PPAM; and the Bloch wavenumber is given by the expression $\cos K\Lambda^p = \mathrm{Re}A$.

The reflection coefficient for the cholesteric has the following form [32]:

$$r_L^c = \frac{i\delta\sin\beta L}{[\beta q/\kappa^2]\cos\beta L + i\left[(q/2\kappa)^2 + (\beta/\kappa)^2 - 1\right]\sin\beta L}, \tag{12}$$

$$\arg r_L^c = \varphi^c. \tag{13}$$

where $\beta = \kappa\left[1 + (q/2\kappa)^2 - \left[(q/\kappa)^2 + \delta^2\right]^{1/2}\right]^{1/2}$ and $\kappa = \omega\varepsilon_m/c$.

The eigenfrequency $\omega_{COTS_1}$ is presented in Figures 3a and 4a as red circles. The analytical solutions of Equation (8) are presented as black circles. The analytical solutions for coupled COTS modes obtained by the coupled oscillator model according to Equation (4) are also presented in Figures 3a and 4a as magenta circles. This solution explains the avoided crossing behavior due to coupling between two modes. It should be noted that the solution obtained by the coupled oscillator model has a good agreement with the simulation results.

### 3.2. Hybrid COTS-CTPP Modes

To demonstrate the hybridization of the chiral optical Tamm state with the chiral Tamm plasmon polariton, we replace the cholesteric with a metasurface (see Figure 5). In this case, the COTS is excited at the CLC-PPAM interface, while the chiral Tamm plasmon polariton is excited at the PPAM-metasurface interface. The fabrication process, scanning electron microscopic image, and spectral measurements for similar metasurfaces were described in detail in our previous publication [36]. In this work, the influence of the angle between the optical axis of the PPAM and the metasurface on the spectral position of the resonance lines has already been studied, but the influence of the period of the metasurface on the possibility of the formation of coupled modes has not been considered. In this regard, we have calculated the reflection spectra of the structure for different values of the metasurface period $p_{ms}$. The results are shown in Figure 6a.

It can be seen from the figure that a smooth increase in the metasurface period leads to an avoided crossing of the resonant lines. For a metasurface period equal to 344 nm, the COTS resonant line is collapsed (see Figure 6b), in other words, the COTS becomes a bound state in the continuum. The mechanism of formation of a bound state in the continuum can be explained by the destructive interference of two coupled modes. For the first time, such a mechanism of interaction was proposed by Friedrich and Wintgen in their work. In the proposed structure, destructive interference becomes possible when the phase difference

between the wave reflected from the PPAM and the waves passed through the PPAM, reflected from the metasurface, and re-passed through the PPAM is equal to $\pi$:

$$2\varphi_t^p + \varphi^{ms} + \psi = \pi, \tag{14}$$

where $\varphi_t^p$ is the PPAM transmission phase, $\varphi^{ms}$ is reflection phase from metasurface, and $\psi$ is the geometric phase at the PPAM-metasurface interface. It should be noted that, in our case, $\psi = \pi/2$.

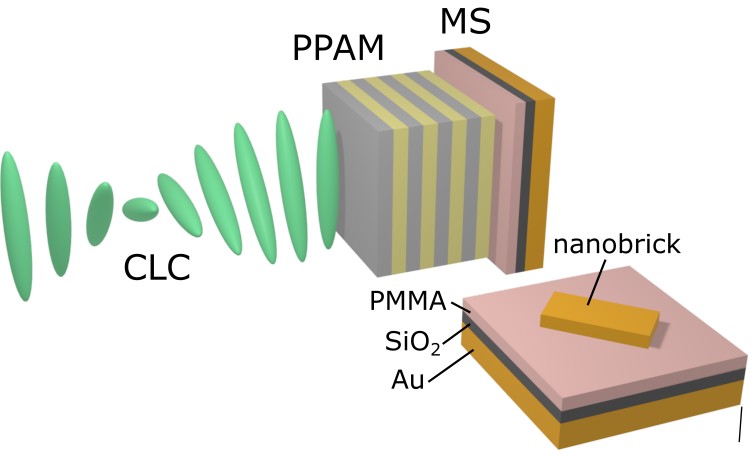

**Figure 5.** Sketch view of the CLC-PPAM-metasurface (MS) structure. The parameters of the PPAM and CLC are the same as Figure 1. The thicknesses of the PMMA, SiO$_2$, and Au substrate are 70 nm, 100 nm, and 200 nm, respectively. Length, width, and height of the nanobrick are 190 nm, 70 nm, and 70 nm, respectively. The period of the metasurface is $p_{ms}$.

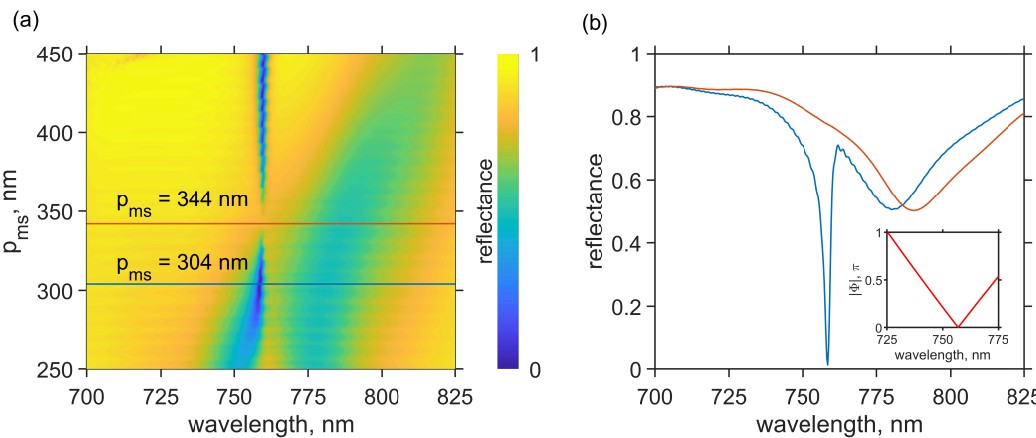

**Figure 6.** (**a**) Reflectance spectra of the CLC-PPAM-MS structure calculated using FDTD method for different values of the metasurface period. (**b**) Reflectance spectra of the structure for metasurface periods of 304 nm and 344 nm. The inset shows a graphical solution of Equation (14).

These phases were calculated using the FDTD method. The simulation results are presented in the inset of Figure 6b. For the convenience of perception, we depicted $|\Phi|$ as equal to the sum of all phases normalized by the value of $\pi$. It can be seen from the figure that $|\Phi|$ takes a zero value at a wavelength of 756 nm, which is completely consistent with the results of direct numerical calculation performed by the FDTD method. Thus, the variation in the period of the metasurface allows for obtaining a bound state in the continuum of the Friedrich–Wintgen type.

The reflection minima near the bound state in the continuum point are due to the fulfillment of the critical coupling condition of the incident field with the COTS [57]. The mechanism of critical coupling point formation can be explained using the coupled mode

theory (TCMT). According to this theory, a localized mode has eigenfrequency $\omega_0$ and energy relaxation times $\tau_l$, $l = 1 \ldots N$ in $N$ channels through which energy can flow into or out of a localized mode. In a conventional CLC–PPAM structure, the energy stored in the COTS can escape through three channels. The first channel with relaxation time $\tau_{\mathrm{p}}$ is related to the transmission of light through the finite PPAM. The second channel with relaxation time $\tau_\phi$ is related to the polarization loss at the CLC-PPAM interface. The third channel with relaxation time $\tau_c$ is related to the transmittance of the finite cholesteric layer. In this case, the spectral resonance linewidth corresponding to the COTS is determined by the total rate of energy leakage from the localized mode and can be determined as [38]

$$\frac{1}{\tau} = \frac{1}{\tau_{\mathrm{p}}} + \frac{1}{\tau_\phi} + \frac{1}{\tau_c}, \tag{15}$$

where

$$
\begin{aligned}
\tau^p &= \sqrt{\frac{\bar{\varepsilon}}{\varepsilon_o^p}} q^{-2N} \frac{2\sqrt{\bar{\varepsilon}}\mathcal{L}}{c}, \\
\tau^\phi &= \sin^{-2}(\psi/2) \frac{2\sqrt{\bar{\varepsilon}}\mathcal{L}}{c}, \\
\tau^c &= e^{\frac{4\pi|n_f|L}{\lambda_0}} \frac{2\sqrt{\bar{\varepsilon}}\mathcal{L}}{c}.
\end{aligned}
\tag{16}
$$

Conjugation of the CLC-PPAM structure with the metasurface leads to the appearance of additional terms in Equation (15):

$$\frac{1}{\tau} = \frac{1}{\tau_{\mathrm{a}}} + \frac{1}{\tau_\phi} + \frac{1}{\tau_\psi} + \frac{1}{\tau_c}, \tag{17}$$

where $\tau_{\mathrm{a}}$ is the relaxation time to the metasurface absorption channel and $\tau_\psi$ is the relaxation time to the polarization loss at the PPAM-metasurface interface.

In this case, the critical coupling condition is achieved when comparing the relaxation times to the cholesteric transmission channel with the remaining relaxation times:

$$\frac{1}{\tau_c} = \frac{1}{\tau_{\mathrm{a}}} + \frac{1}{\tau_\phi} + \frac{1}{\tau_\psi}. \tag{18}$$

A gradual increase in the metasurface period leads to the change in its absorption and polarization losses. It can be seen from Figure 6 that the critical coupling condition is achieved at metasurface periods of 300 and 400 nm. This means that for these periods, the relaxation rates to the metasurface absorption and polarization loss channels satisfy Equation (18). Thus, when the critical coupling condition of the incident field with the COTS is achieved, the absorption coefficient goes to unity; in other words, the proposed structure can be used as a perfect absorber.

## 4. Conclusions

The mechanism of hybridization of two chiral optical Tamm modes localized at the interface of the cholesteric—an anisotropic polarization-preserving mirror, as well as the chiral optical Tamm state with a chiral Tamm plasmon polariton arising at the interface of the PPAM—polarization-preserving metasurface is investigated. Both structures allow the hybridization of modes. It is shown that changing the cholesteric pitch, the number of layers, the polarization-preserving anisotropic mirror, the angle between the optical axis of the cholesteric and the PPAM, and also the period of the polarization-preserving metasurface allows you to control the splitting of hybrid modes and to control the strength of their coupling.

However, the proposed structure containing a cholesteric-PPAM-metasurface has a large number of degrees of freedom, which makes it possible to obtain a bound state in

the continuum of the Friedrich–Wintgen type [58]. It was also shown that at the points of critical coupling, the absorption reached one hundred percent, which makes it possible to use such a structure as a perfect absorber.

**Author Contributions:** N.V.R., methodology, visualization, writing—original draft; R.G.B., methodology, visualization; S.Y.V., conceptualization, validation, supervision; L.E.T., methodology; I.V.T., conceptualization, methodology, software, validation, supervision, writing—review and editing. All authors have read and agreed to the published version of the manuscript.

**Funding:** The work was carried out within the state assignment of the Federal Research Center KSC SB RAS.

**Data Availability Statement:** The data presented in this study are available on request from the corresponding author.

**Conflicts of Interest:** The authors declare no conflict of interest.

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
