# Peer review of "Tuning Q-Factor and Perfect Absorption Using Coupled Tamm States on Polarization-Preserving Metasurface"

_photonics, doi:10.3390/photonics10121391_

Round 1

Reviewer 1 Report

Comments and Suggestions for Authors

The paper extends the use of chiral Tamm states by demonstating coupling between such states, and with the extension of coupling to a meta surface. Clearly explained and presented design, will make a clear contribution to the Tamm state literate and be sited by those of us who work on Tamm states and metasurfaces, as well as being of broader interest as of another, subtle example for realizing bound states in the continuum. I would have liked some comment on the challenges inherent in fabricating such devices rather than just simulating them, but the sound conclusions are a valuable contribution to the literature and suitable for Photonics.  

Comments on the Quality of English Language

Very minor misspelling "hybride" for "hybrid" in standard English. 

Author Response

Reply to Reviewer 1

Review Report (Round 1)

The paper extends the use of chiral Tamm states by demonstating coupling between such states, and with the extension of coupling to a meta surface. Clearly explained and presented design, will make a clear contribution to the Tamm state literate and be sited by those of us who work on Tamm states and metasurfaces, as well as being of broader interest as of another, subtle example for realizing bound states in the continuum. I would have liked some comment on the challenges inherent in fabricating such devices rather than just simulating them, but the sound conclusions are a valuable contribution to the literature and suitable for Photonics.

Response: We thank Reviewer 1 for constructive comments that greatly motivate us for the research and request the challenges inherent in fabricating such devices. The cholesteric and metasurface interface was already fabricated experimentally in our previous paper [Lin, M.Y.; Xu, W.H.; Bikbaev, R.G.; Yang, J.H.; Li, C.R.; Timofeev, I.V.; Lee, W.; Chen, K.P. Chiral-Selective Tamm Plasmon Polaritons. Materials 2021, 14.]. The possible way of fabrication of the newly-proposed structure is to use polymer-stabilized liquid crystals. Polarization-preserving anisotropic mirror (PPAM) can be made of polymerized nematic layers, for example 5CB nematic. Each new layer of LC should be applied sequentially and polymerized. The main challenge in fabricating such devices is to keep plain parallel boundaries in a series of layers.

Comments on the Quality of English Language

Very minor misspelling "hybride" for "hybrid" in standard English.

Response: We thank Reviewer 1 for this comment. The improvements in English are marked in the updated manuscript.

Reviewer 2 Report

Comments and Suggestions for Authors

In this manuscript, the authors studied coupled modes formed as a result of the interaction of two chiral optical Tamm states or a chiral optical Tamm state and a chiral Tamm plasmon polariton. It is shown that effective control of coupled modes can be carried out by changing the pitch of the cholesteric and the angle between the optical axis of the cholesteric and the polarization-preserving anisotropic mirror. The influence of the metasurface period on the spectral characteristics of coupled modes is investigated. The possibility of realizing a bound state in a continuum of the Friedrich-Wintgen type, resulting from the destructive interference of coupled modes, which leads to the collapse of the resonance line corresponding to the chiral optical Tamm state, has been demonstrated. The structure is clear and the figures are prepared in high quality. I would like to recommend it to be published in Photonics if the authors can address my following comments:

1. For formula 4, the author can further provide an expression for the resonance frequency. Related expressions can be found in reference [Journal of Physics D: Applied Physics 52, 015104 (2019)].

2. To help readers better understand the Tamm modes, the authors can provide relevant field distributions.

3. Regarding the schematic diagram of Figure 5, considering that there is nanobrick at the top of the MS structure, should there be a gap between PPAM and MS, which is the height of the nanobrick?

4. Some related references are missed, such as [Optics & Laser Technology 159 (2023): 108928.], [Optics Letters 44 (13), 3302-3305 (2019)], [ACS Photonics, 2016, 3(10): 1776-1781.], etc.

Author Response

Reply to Reviewer 2

Review Report (Round 1)

In this manuscript, the authors studied coupled modes formed as a result of the interaction of two chiral optical Tamm states or a chiral optical Tamm state and a chiral Tamm plasmon polariton. It is shown that effective control of coupled modes can be carried out by changing the pitch of the cholesteric and the angle between the optical axis of the cholesteric and the polarization-preserving anisotropic mirror. The influence of the metasurface period on the spectral characteristics of coupled modes is investigated. The possibility of realizing a bound state in a continuum of the Friedrich-Wintgen type, resulting from the destructive interference of coupled modes, which leads to the collapse of the resonance line corresponding to the chiral optical Tamm state, has been demonstrated. The structure is clear and the figures are prepared in high quality. I would like to recommend it to be published in Photonics if the authors can address my following comments:

  1. For formula 4, the author can further provide an expression for the resonance frequency. Related expressions can be found in reference [Journal of Physics D: Applied Physics 52, 015104 (2019)].

Response: We thank the Reviewer for this comment.

Expression for the resonance frequency has been added to the text. The reference [Journal of Physics D: Applied Physics 52, 015104 (2019)] has been added also.

“As a result the resonant frequencies of the modes can be expressed as [54]:

<Please, see the attached pdf-file>

  1. Qing, Y.M.; Ma, H.F.; Yu, S.; Cui, T.J. Tunable dual-band perfect metamaterial absorber based on a graphene-SiC hybrid system by multiple resonance modes. Journal of Physics D: Applied Physics 2018, 52, 015104. https://doi.org/10.1088/1361-6463/aae75f.”
  2. To help readers better understand the Tamm modes, the authors can provide relevant field distributions.

Response: The field distribution at the COTS wavelength has been added to Figure 2c.

“Figure 2. (c) Field distribution at the COTS wavelength λ = 750 nm.”

  1. Regarding the schematic diagram of Figure 5, considering that there is nanobrick at the top of the MS structure, should there be a gap between PPAM and MS, which is the height of the nanobrick?

Response: We thank the Reviewer for this comment. This inaccuracy has been corrected. A gap between PPAM and MS has been added to Figure 5.

  1. Some related references are missed, such as [Optics & Laser Technology 159 (2023): 108928.], [Optics Letters 44 (13), 3302-3305 (2019)], [ACS Photonics, 2016, 3(10): 1776-1781.], etc.

Response: The reference has been added to the text.

“6. Kar, C.; Jena, S.; Udupa, D.V.; Rao, K.D. Tamm plasmon polariton in planar structures: A briefoverview and applications. Optics Laser Technology 2023, 159, 108928. https://doi.org/10.1016/j.optlastec.2022.108928.

  1. Azzini, S.; Lheureux, G.; Symonds, C.; Benoit, J.M.; Senellart, P.; Lemaitre, A.; Greffet, J.J.; Blanchard, C.; Sauvan, C.; Bellessa, J. Generation and Spatial Control of Hybrid Tamm Plasmon/Surface Plasmon Modes. ACS Photonics 2016, 3, 1776–1781. https://doi.org/10.1021/acsphotonics.6b00521.
  2. Qing, Y.M.; Ma, H.F.; Cui, T.J. Flexible control of light trapping and localization in a hybrid Tamm plasmonic system. Opt. Lett. 2019, 44, 3302–3305. https://doi.org/10.1364/OL.44.003302.”

Reviewer 3 Report

Comments and Suggestions for Authors

Comments on the manuscript

The manuscript explores the tuning of Q-factor and perfect absorption through coupled Tamm states on a polarization-preserving metasurface. It investigates the interaction of chiral optical Tamm states and chiral Tamm plasmon polaritons, demonstrating effective control by varying parameters like cholesteric pitch, layers of the anisotropic mirror, and the metasurface period. The study reveals the potential for a bound state in a continuum, leading to the collapse of the resonance line corresponding to the chiral optical Tamm state. The proposed structure, with its numerous degrees of freedom, allows for critical coupling points, achieving 100% absorption, making it a promising perfect absorber for various applications.

In my assessment, the manuscript is interesting, and the theoretical/simulational findings contribute to plasmonic BIC absorbers, a topic of high interest. Overall, the manuscript is methodologically robust, with well-supported claims and conclusions. Consequently, this manuscript aligns with the scope of the Photonics, pending the resolution of a few minor concerns. Please find my suggestions and comments for the authors below.

Some typos. “…of the interaction of different types of modes is the hybrid state of the TPP and exciton polaritons [35? ,36], the…” an unnecessary question.

"The reflection minima near the bound state in the continuum point are due to the fulfillment of the critical coupling condition of the incident field with the COTS." Correct. However, some works on plasmonic perfect BIC absorbers at critical conditions are missing, which can support your claim. For example, ["Toroidal Dipole BIC-Driven Highly Robust Perfect Absorption with a Graphene-Loaded Metasurface." Nano Letters 23.19 (2023): 9105-9113; "Bound states in the continuum in anisotropic plasmonic metasurfaces." Nano Letters 20.9 (2020): 6351-6356.]

What is the significance of observing only one resonant line in the reflectance spectra when N > 20 in Figure 2, and how does this behavior relate to the coupled COTS modes?

How does the helix pitch of the CLC affect the splitting value of the coupled COTS modes, as evidenced by the reflectance spectra for different CLC pitch values in Figure 3?

At what helix pitch of the CLC is the minimum splitting value achieved, as shown in Figure 3, and what are the implications of this optimal pitch in practical applications?

Comments on the Quality of English Language

readable

Author Response

Reply to Reviewer 3

Review Report (Round 1)

The manuscript explores the tuning of Q-factor and perfect absorption through coupled Tamm states on a polarization-preserving metasurface. It investigates the interaction of chiral optical Tamm states and chiral Tamm plasmon polaritons, demonstrating effective control by varying parameters like cholesteric pitch, layers of the anisotropic mirror, and the metasurface period. The study reveals the potential for a bound state in a continuum, leading to the collapse of the resonance line corresponding to the chiral optical Tamm state. The proposed structure, with its numerous degrees of freedom, allows for critical coupling points, achieving 100% absorption, making it a promising perfect absorber for various applications.

In my assessment, the manuscript is interesting, and the theoretical/simulational findings contribute to plasmonic BIC absorbers, a topic of high interest. Overall, the manuscript is methodologically robust, with well-supported claims and conclusions. Consequently, this manuscript aligns with the scope of the Photonics, pending the resolution of a few minor concerns. Please find my suggestions and comments for the authors below.

Response: We thank Reviewer 3 for his positive evaluation of our work.

Comment 1: Some typos. “…of the interaction of different types of modes is the hybrid state of the TPP and exciton polaritons [35? ,36], the…” an unnecessary question.

Response:  This typo has been corrected.

Comment 2: "The reflection minima near the bound state in the continuum point are due to the fulfillment of the critical coupling condition of the incident field with the COTS." Correct. However, some works on plasmonic perfect BIC absorbers at critical conditions are missing, which can support your claim. For example, ["Toroidal Dipole BIC-Driven Highly Robust Perfect Absorption with a Graphene-Loaded Metasurface." Nano Letters 23.19 (2023): 9105-9113; "Bound states in the continuum in anisotropic plasmonic metasurfaces." Nano Letters 20.9 (2020): 6351-6356.]

Response: We thank Reviewer 3 for his constructive comment. The references have been added to the text:

“59. Liang, Y.; Koshelev, K.; Zhang, F.; Lin, H.; Lin, S.; Wu, J.; Jia, B.; Kivshar, Y. Bound States in the Continuum in Anisotropic Plasmonic Metasurfaces. Nano Letters 2020, 20, 6351–6356. PMID: 32479094, https://doi.org/10.1021/acs.nanolett.0c01752.

  1. Jin, R.; Huang, L.; Zhou, C.; Guo, J.; Fu, Z.; Chen, J.; Wang, J.; Li, X.; Yu, F.; Chen, J.; et al. Toroidal Dipole BIC-Driven Highly Robust Perfect Absorption with a Graphene-Loaded Metasurface.Nano Letters 2023, 23, 9105–9113. PMID: 37694889, https://doi.org/10.1021/acs.nanolett.3c02958.”

Comment 3: What is the significance of observing only one resonant line in the reflectance spectra when N > 20 in Figure 2, and how does this behavior relate to the coupled COTS modes?

Response: The reason for observing only one resonant mode at N = 20 is that the coupling between modes is weak, since the fields localized at the CLC-PPAM and PPAM-CLC interfaces do not overlap. This is clearly seen from Figure 2c. A decrease in the number of PPAM layers leads to overlapping fields and removal of degeneracy. As a consequence, two resonant lines appear in the reflection spectrum instead of one. This effect is described in sufficient detail in the text:

“A decrease in the PPAM periods number leads to a decrease in reflection within its band gap and, as a consequence, the incident radiation tunnels through it to a second interface with CLC. As a result, a second COTS is excited at the PPAM-CLC boundary. The states localized at the PPAM boundaries begin to overlap with each other, which leads to removal of degeneracy and splitting of the resonant line.”

Comment 4: How does the helix pitch of the CLC affect the splitting value of the coupled COTS modes, as evidenced by the reflectance spectra for different CLC pitch values in Figure 3?

Response: The splitting value is adequately described by new eq. (5):

<Please, see the attached pdf-file>

Please, also see our response to the next comment.

Comment 5: At what helix pitch of the CLC is the minimum splitting value achieved, as shown in Figure 3, and what are the implications of this optimal pitch in practical applications?

Response:  Additional explanations about splitting value have been added to the text: ”The minimum splitting value is achieved at a helix pitch of 750 nm, while a decrease or increase in this value leads to an increase in the splitting value.”

The implication of this optimal pitch in practical applications is its tunability. Indeed we can control on demand the distance and the minimal distance by variation of the CLC pitch and number of PPAM layers.

Reviewer 4 Report

Comments and Suggestions for Authors

In my opinion, the work is addressed to a narrow group of readers. In particular, theoretical physicists working in the field of photonics. It's easy to feel it when reading the abstract. Of course, such a presentation of research results is acceptable, the question is whether it is optimal. Because, it would be useful to provide information for a wider circle of potentially interested recipients what the authors actually want. As temporal coupled-mode theory is known but not as popular as authors may think. As a aim of work I can consider “The possibility of realizing a bound state in a continuum of the Friedrich-Wintgen type, resulting from the destructive interference of coupled modes, which leads to the collapse of the resonance line corresponding to the chiral optical Tamm state, has been demonstrated.” But I have the soul of an engineer and I would like to know what comes out of it apart from a scientific publication.

These are some thoughts on the abstract. In the "introduction" part it is much better. Nevertheless, I kindly ask you not to use multi-letter abbreviations in drawings without presenting a legend on figure 1. Figure 1: Is the structure hanging in the air? What is the difference between clc1 and clc2, because in the following part of the publication the symmetrical structure from the point of view of clc is presented, but symmetry does not apply to the structure of ppam? Is this what the authors meant? Because the presented tensors look symmetric.

I am in sensing devices design, construction, preparation and data analysis including ANN for 30 years and I'm not convinced by “can be used for intelligent design of laser and sensor devices”. I do not know any sensing device that use on 8-9 TRL such structures, and I do not know what is intelligent design of laser and sensor devices. Please give precise examples and references including documentation of such devices and methods or change the meaning of sentence.

Is the work only about modeling or is there any part of experimental verification as well?

It would be very useful to show the possibility of practical tuning Q-factor of the laser with the modeled structure use.

Author Response

Reply to Reviewer 4

Review Report (Round 1)

In my opinion, the work is addressed to a narrow group of readers. In particular, theoretical physicists working in the field of photonics. It's easy to feel it when reading the abstract. Of course, such a presentation of research results is acceptable, the question is whether it is optimal. Because, it would be useful to provide information for a wider circle of potentially interested recipients what the authors actually want. As temporal coupled-mode theory is known but not as popular as authors may think. As a aim of work I can consider “The possibility of realizing a bound state in a continuum of the Friedrich-Wintgen type, resulting from the destructive interference of coupled modes, which leads to the collapse of the resonance line corresponding to the chiral optical Tamm state, has been demonstrated.” But I have the soul of an engineer and I would like to know what comes out of it apart from a scientific publication.

Response: We thank Reviewer 4 for his opinion.

Comment 1: These are some thoughts on the abstract. In the "introduction" part it is much better. Nevertheless, I kindly ask you not to use multi-letter abbreviations in drawings without presenting a legend on figure 1. Figure 1: Is the structure hanging in the air? What is the difference between clc1 and clc2, because in the following part of the publication the symmetrical structure from the point of view of clc is presented, but symmetry does not apply to the structure of ppam? Is this what the authors meant? Because the presented tensors look symmetric.

Response: The caption of the Figure 1 has been reworked. Additional explanations about structure have been added to the text: “CLC1 and CLC2 are denoted cholesterics with a different helix pitch. The investigated structure is placed in media with refractive index $n_av = (n_o+n_e)/2$.”

Comment 2: I am in sensing devices design, construction, preparation and data analysis including ANN (artificial neural network) for 30 years and I'm not convinced by “can be used for intelligent design of laser and sensor devices”. I do not know any sensing device that use on 8-9 TRL (technology readiness level: 8 near-commercialization & 9 commercialization) such structures, and I do not know what is intelligent design of laser and sensor devices. Please give precise examples and references including documentation of such devices and methods or change the meaning of sentence.

The reviewer is right on sensing devices on 8-9 technology readiness level. In scientific journals it is conventional to talk on TRL 1 (Basic principles observed and reported). Indeed, there are several TRL 1 laser and sensor devices based on TPP reported. Among them TPP-VCSELs [Qiao P., Yang W., and  Chang-Hasnain C., Recent advances in high-contrast metastructures, metasurfaces, and photonic crystals, Adv. Opt. Photon. 10, 180-245 (2018).] and PCSELs [ Pan C.,  Lin C.,  Chang T.,  Lu T., and  Lee C., "GaSb-based mid infrared photonic crystal surface emitting lasers," Opt. Express 23, 11741-11747 (2015).], CLC-DBR lasers [Huang, JC., Hsiao, YC., Lin, YT. et al. Electrically switchable organo–inorganic hybrid for a white-light laser source. Sci Rep 6, 28363 (2016). https://doi.org/10.1038/srep28363]. Recently, chiral TPP was experimentally observed and proposed as a temperature sensor [Lin, M.Y.; Xu, W.H.; Bikbaev, R.G.; Yang, J.H.; Li, C.R.; Timofeev, I.V.; Lee, W.; Chen, K.P. Chiral-Selective Tamm Plasmon Polaritons. Materials 2021, 14.]. The text was modified to avoid possible equivocal meaning:

“... especially vertical-cavity lasers [26, 27] and…. "

“The synergy of active cholesteric and reflective PC yields CLC-PC lasers [32]. The localized surface state that is excited at the interface between a CLC and a chiral plasmonic mirror, exponentially decreases with the increasing distance from the interface. It is called a chiral Tamm plasmon polariton (CTPP) [2]. Recently, it was experimentally investigated and proposed as a temperature sensor device [33]. The Q-factor can be improved by replacing the plasmonic metasurfaces with a dielectric polarization-preserving anisotropic mirror (PPAM) [34]. As a result, a high-quality chiral optical Tamm state (COTS) [35] is excited at the CLC-PPAM interface. The spectral manifestation of COTS can be observed both upon reflection and transmission. The high reflectance coefficient from the PPAM is provided by a large number of layers, which complicated the technology of manufacturing of this mirrors. Reducing of the PPAM layers, while maintaining the Q-factor of the localized state, can be achieved due to its conjugation with the metasurface, as it was demonstrated in the work [36].”

Comment 3: Is the work only about modeling or is there any part of experimental verification as well? It would be very useful to show the possibility of practical tuning Q-factor of the laser with the modeled structure use.

Response: We thank the Reviewer for this comment. Yes, this article presents only modeling. Practical tuning Q-factor of the laser with the modeled structure is shown in our separate work [G. A. Romanenko, P. S. Pankin, D. S. Buzin, D. N. Maksimov, V. S. Sutormin, A. I. Krasnov, F. V. Zelenov, A. N. Masyugin, S. V. Nedelin, N. A. Zolotovskiy, I. A. Tambasov, M. N. Volochaev, K.-P. Chen, I. V. Timofeev; Metal–dielectric optical microcavity with tunable Q factor. Appl. Phys. Lett. 7 August 2023; 123 (6): 061113. https://doi.org/10.1063/5.0157430]. The annotation was modified to stress this:

“In this report coupled modes formed as a result of the interaction of two chiral optical Tamm states or a chiral optical Tamm state and a chiral Tamm plasmon polariton are analytically and numerically investigated.”
